# NOT-SO-CLEVR: VISUAL RELATIONS STRAIN FEED-FORWARD NEURAL NETWORKS

## ABSTRACT

The robust and efficient recognition of visual relations in images is a hallmark of biological vision. Here, we argue that, despite recent progress in visual recognition, modern machine vision algorithms are severely limited in their ability to learn visual relations. Through controlled experiments, we demonstrate that visual-relation problems strain convolutional neural networks (CNNs). The networks eventually break altogether when rote memorization becomes impossible such as when the intra-class variability exceeds their capacity. We further show that another type of feedforward network, called a relational network (RN), which was shown to successfully solve seemingly difficult visual question answering (VQA) problems on the CLEVR datasets, suffers similar limitations. Motivated by the comparable success of biological vision, we argue that feedback mechanisms including working memory and attention are the key computational components underlying abstract visual reasoning.

## 1 INTRODUCTION

Consider the two images in Fig. 1. The image on the left was correctly classified as a flute by a deep convolutional neural network (CNN) (He et al., 2015). This is quite a remarkable feat for such a complicated image, which includes distractors that partially occlude the object of interest. After the network was trained on millions of photographs, this and many other images were accurately categorized into one thousand natural object categories, surpassing, for the first time, the accuracy of a human observer on the ImageNet classification challenge.

Now, consider the image on the right. On its face, it is quite simple compared to the image on the left. It is just a binary image containing two curves. Further, it has a rather distinguishing property, at least to the human eye: both curves are the same. The relation between the two items in this simple scene is rather intuitive and immediately obvious to a human observer. Yet, the CNN failed to learn this relation even after seeing millions of training examples.

Why is it that a CNN can accurately detect the flute in Fig. 1a while struggling to recognize the simple relation depicted in Fig. 1b?

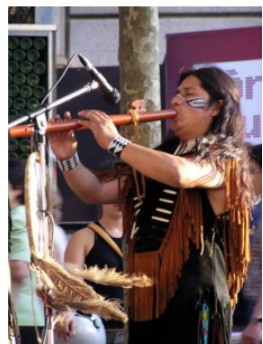 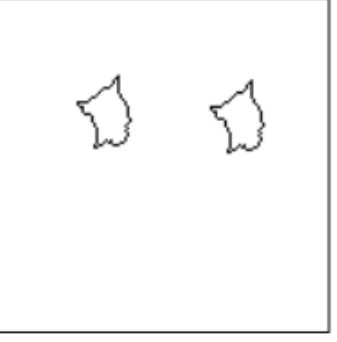

(a) (b)

Figure 1: *Two images:* The image in panel *(a)* can be classified with high confidence as containing a flute by contemporary computer vision algorithms. However, these same algorithms struggle to learn the concept of "sameness" as exemplified by the image with the two curves shown in panel *(b)*. The image in panel *(b)* is sampled from the SVRT challenge (Fleuret et al., 2011).

That such task is difficult, and even sometimes impossible for contemporary computer vision algorithms including CNNs, is known (Fleuret et al., 2011; Gülçehre and Bengio, 2013; Ellis et al., 2015; Stabinger et al., 2016) but has, so far, been overlooked. To make matters worse, the issue has been overshadowed by the recent success of a novel class of neural networks called relational networks (RNs) on seemingly challenging visual question answering (VQA) benchmarks. However, RNs have so far only been tested using toy datasets like the sort-of-CLEVR dataset which depicts combinations of items of only a handful of colors and shapes (Santoro et al., 2017). As we will show, RNs suffer the same limitations as CNNs for a same-different task such as the one shown in Fig. 1b.

This failure of modern computer vision algorithms is all the more striking given the widespread ability to recognize visual relations across the animal kingdom, from human and non-human primates (Donderi and Zelnicker, 1969; Katz and Wirght, 2006) to rodents (Wasserman et al., 2012), birds (Daniel et al., 2015; Martinho III and Kacelnik, 2016) and even insects (Giurfa et al., 2001). Examining the failures of existing models is a critical step on the path to understanding the computational principles underlying visual reasoning. Yet, to our knowledge, there has not been any systematic exploration of the limits of contemporary machine learning algorithms on relational reasoning problems.

Previous work by Fleuret et al. (2011) showed that black-box classifiers fail on most tasks from the synthetic visual reasoning test (SVRT), a battery of twenty-three visual-relation problems, despite massive amounts of training data. More recent work by Ellis et al. (2015) and Stabinger et al. (2016) each showed how two different CNN architectures could only solve a handful of the twenty-three SVRT problems. Similarly, Gülçehre and Bengio (2013), after showing how CNNs fail to learn a same-different task with simple binary "sprite" items, only managed to train a multi-layer perceptron on this task by providing carefully engineered training schedules. However, these results of Ellis et al. (2015), Gülçehre and Bengio (2013) and Stabinger et al. (2016) were inconclusive: does the inability of feedforward neural networks to solve various visual-relation problems reflect a poor choice of hyperparameters for their particular implementation or rather a systematic failure of the entire class of feedforward models?

Here, we propose to systematically probe the limits of CNNs and other state-of-the-art visual reasoning networks (RNs) on visual-relation tasks. Through a series of controlled experiments, we demonstrate that visual-relation tasks strain CNNs and that these limitations are not alleviated in RNs, which were specifically designed to tackle visual-relation problems. A brief review of the biological vision literature suggests that two key brain mechanisms, working memory and attention, underlie primates' ability to reason about visual relations. We argue that these mechanisms and possibly other feedback mechanisms are needed to extend current computer vision models to efficiently learn to solve complex visual reasoning tasks.

Our contributions are threefold: (i) We perform the first systematic performance analysis of CNN architectures on each of the twenty-three SVRT problems. This yields a dichotomy of visual-relation problems, hard same-different problems vs. easy spatial-relation problems. (ii) We describe a novel, controlled, visual-relation challenge which convincingly shows that CNNs solve same-different tasks via rote memorization. (iii) We show that a simple modification of the sort-of-CLEVR challenge similarly breaks state-of-the-art relational network architectures.

Overall, we wish to motivate the computer vision community to reconsider existing visual question answering challenges and turn to neuroscience and cognitive science for inspiration to help with the design of visual reasoning architectures.

## 2   EXPERIMENT 1: SVRT

**The SVRT challenge:**   This challenge is a collection of twenty-three binary classification problems in which opposing classes differ based on whether their stimuli obey an abstract rule (Fleuret et al., 2011). For example, in problem number 1, positive examples feature two items which are the same up to translation, whereas negative examples do not (Fig. 2a). In problem 9, positive examples have three items, the largest of which is in between the two smaller ones (Fig. 2b). All stimuli depict simple, closed, black curves on a white background.

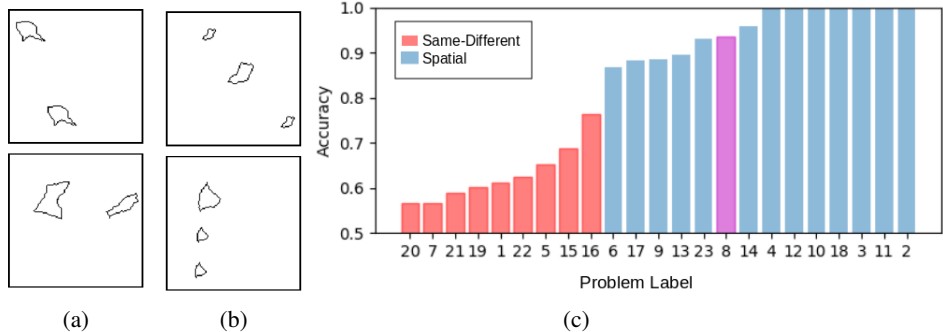

(a)  (b)  (c)

Figure 2: *The Synthetic Visual Reasoning Test. (a)* and *(b)* show an example pair of Problem 1 (*same-different up to translation*) and Problem 2 (*three objects arranged in a line with the largest one in the middle*), respectively. *(c)* Nine CNNs corresponding to different combinations of hyper-parameters were trained on each of the twenty-three SVRT problems. Shown are the ranked accuracies of the best-performing network for each problem. The $x$-axis indicates the arbitrary problem label provided in (Fleuret et al., 2011). CNNs from this high-throughput analysis were found to produce uniformly lower accuracies on same-different problems (red bars) than on spatial-relation problems (blue bars). The single purple bar corresponds to a problem which required detecting both a same-different relation and a spatial relation simultaneously.

**High-throughput screening approach:** We tested CNNs of three different depths (2, 4 and 6 convolutional layers) and three different convolutional receptive field sizes (2×2, 4×4 and 6×6) for a total of nine networks. All networks used pooling kernels of size 3×3, convolutional strides of 1, pooling strides of 2 and three fully connected layers. Pooling layers used ReLu activations. For each of the twenty-three problems, we generated 2 million examples split evenly into training and test sets using code publicly provided by Fleuret et al. (2011) at `http://www.idiap.ch/~fleuret/svrt/`. We trained all nine networks on each problem for a total of $n = 207$ conditions. All networks were trained using an ADAM optimizer with base learning rate of $\eta = 10^{-4}$.

**Results:** The accuracy of the best networks obtained for each problem individually (across all networks) is shown in Fig. 2c. After the best-case performance for each of the twenty-three problems was obtained, we sorted the problems by accuracy and then colored the bars red or blue according to the SVRT problem descriptions provided by (Fleuret et al., 2011). Problems whose descriptions have words like "same" or "identical" are colored red. These *Same-Different* (SD) problems have items that are congruent up to some transformation (e.g., Problem 1 in Fig. 2a). *Spatial-Relation* (SR) problems, whose descriptions have phrases like "left of", "next to" or "touching," are colored blue.

The resulting dichotomy across the SVRT problems is striking. Evident from Fig. 2c is the fact that CNNs fare much worse on SD problems than they do on SR problems. Many SR problems were learned satisfactorily, whereas some SD problems (e.g., problems 20, 7 and 21) resulted in accuracy not substantially above chance. From this analysis, it appears as if SD tasks pose a particularly difficult challenge to CNNs. This result matches earlier evidence for a visual-relation dicohtomy provided by Stabinger et al. (2016) which was unknown to us at the time of our own experiments. Additionally, our hyperparameter search revealed that SR problems are generally equally well-learned across all network configurations, with less than 10% difference in final accuracy between the worst-case and the best-case. On the other hand, larger networks generally yielded significantly higher accuracy on SD problems than smaller ones. If only the results from a single architecture had been reported, the visible dichotomy would have been stronger. Experiment 1 corroborates the results of previous studies which found feedforward models performed badly on many visual-relation problems (Fleuret et al., 2011; Gülçehre and Bengio, 2013; Ellis et al., 2015; Santoro et al., 2017) and suggests that low performance cannot be simply attributed to a poor choice of hyperparameters.

**Limitations of the SVRT challenge:** While the SVRT challenge is useful for surveying the efficacy of an algorithm on a diverse range of visual relations, it has two important limitations. First, the twenty-three problems used in the challenge constitute a somewhat arbitrary sample from a very large set of all conceivable visual relations. While there are some obvious connections between

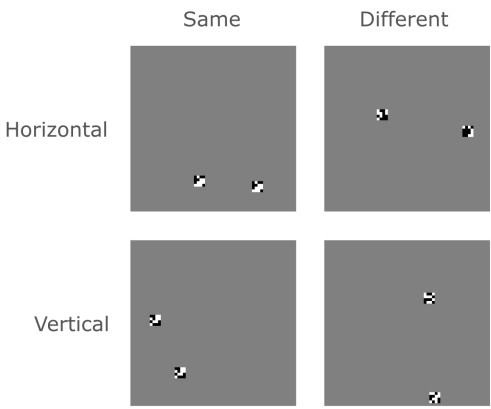

Figure 3: *Sample PSVRT images*. Four images are shown representing the four joint categories of SD (grouped by columns) and SR (grouped by rows). An image is considered *Same* or *Different* depending on whether it contains identical (left column) or different (right column) square bit patterns. An image is considered *Horizontal* (top row) or *Vertical* (bottom row) depending on whether the orientation of the displacement between the items is greater than $45°$. The images were generated with the baseline image parameters: $m = 4$, $n = 60$, $k = 2$.

different problems (e.g., *"same-different up to translation"* in Problem 1 and *"same-different up to scale"* in Problem 19), a direct comparison between most problems is generally hard because they often assume different image structures, each requiring unique image generation methods resulting in different image distributions. For example, Problem 2 (*"inside-outside"*) requires that an image contains one large object and one small object. This necessary configuration naturally conflicts with other problems such as Problem 1 (*"same-different up to translation"*) where two items must be identically-sized and positioned without one being contained in the other. In other cases, problems simply require different number of objects in a single image (two items in Problem 1 vs. three in Problem 9). Instead, a better way to compare visual-relation problems would be to define various problems on the *same* set of images.

Second, using simple, closed curves as items in SVRT images makes it difficult to quantify and control image variability as a function of image generation parameters. While closed curves are perceptually interesting objects, the ad hoc procedure used to generate them prevents quantification of image variability and its effect on task difficulty. As a result, even within a single problem in SVRT, it is unclear whether its difficulty is inherent to the classification rule itself or simply results from the particular choice of image generation parameters unrelated to the rule.

## 3 EXPERIMENT 2: PSVRT

**The PSVRT challenge:** To address the issues associated with SVRT, we constructed a new dataset consisting of two idealized problems from the dichotomy that emerged from Experiment 1 (Fig. 3): *Spatial Relations* (SR) and *Same-Different* (SD). In SR, an image is classified according to whether the items in an image are arranged horizontally or vertically as measured by the orientation of the line joining their centers (with a $45°$ threshold). In SD, an image is classified according to whether or not it contains identical items. As long as we limit the problems to these two simple rules, the same image dataset can be used in both problems by simply labeling each image according to different rules (Fig. 3).

Our image generator produces a gray-scale image by placing square binary bit patterns (consisting of values $1$ and $-1$) on a blank background (with value $0$). The generator uses three parameters to control image variability: item size, image size and number of items in a single image. Item size ($m$) refers to the side-length of the square bit patterns and controls image variability at the item level. Image size ($n$) refers to side-length of the input image. It thus controls image variability by setting the spatial extent of the placement of individual items. Lastly, the number of items ($k$) controls both item and spatial variability since adding one more item in the image increase the total number of possible images by a factor equal to the number of different bit patterns for the new item times the number of positions at which it can be placed.

When $k \geq 3$, the SD category label is determined by whether or not there are *at least 2* identical items in the image, and the SR category label is determined according to whether the *average* orientation of the displacements between all pairs of items is greater than or equal to $45°$. These parameters allowed us to quantify the number of possible images in a dataset as $\mathcal{O}(P_{n^2,k}\, 2^{km^2})$, where

$P_{a,b}$ denotes the number of possible permutations of $a$ elements from a set of size $b$. To highlight the parametric nature of the image samples, we call this test *Parametric* SVRT, or PSVRT.

Each image is generated by first drawing a joint class label for SD and SR from a uniform distribution over $\{Different, Same\} \times \{Horizontal, Vertical\}$. The first item is sampled from a uniform distribution in $\{-1, 1\}^{m \times m}$. Then, if the sampled SD label is *Same*, between 1 and $k - 1$ identical copies of the first item are created. If the sampled SD label is *Different*, no identical copies are made. The rest of $k$ unique items are then consecutively sampled. These $k$ items are then randomly placed in an $n \times n$ image while ensuring at least 1 background pixel spacing between items. Generating images by always drawing class labels for both problems ensures that the image distribution is identical between the two problem types.

**Method and architecture details:** Our goal in this experiment was to examine the difficulty of learning PSVRT problems over a range of image variability parameters. First, we found a baseline architecture which could easily learn both same-different and spatial-relation PSVRT problems for one parameter configuration (item size $m = 4$, image size $n = 60$ and item number $k = 2$). Then, for each combination of item size, image size and item number, we trained an instance of this architecture from scratch. In each training session, we measured the number of training examples required for the architecture to reach $95\%$ accuracy (training-to-acquisition or TTA). We use TTA as a measure of problem difficulty. In all conditions, the network was trained from scratch and training accuracy was measured. We simply wanted to estimate the difficulty of fitting the training data in various conditions (as opposed to generalization). Hence, there was no holdout test set.

We varied each image parameter separately to examine its effect on learnability. This resulted in three sub-experiments:

1. $n$ was varied between 30 and 180 while $m$ and $k$ were fixed at 4 and 2, respectively
2. $m$ was varied between 3 and 7, while $n$ and $k$ were fixed at 60 and 2, respectively
3. $k$ was varied between 2 and 6 while $n$ and $m$ were fixed at 60 and 4, respectively

An instance of the baseline CNN was trained from scratch in each condition with 20 million training images with a batch size of 50. Throughout, we only report the best-case result (minimum TTA) for each experimental condition out of 10 random initializations.

The baseline convolutional network had four convolution and pool layers and four fully-connected layers. The first convolution layer had 16 kernels of size $4 \times 4$, followed by 32, 64 and 128 kernels of size $2 \times 2$ in the subsequent convolution layers. Four pool layers were interleaved after each convolution step with a kernel of size $3 \times 3$ and with strides of size $2 \times 2$. The fully-connected layers had 256 units in each layer. We used dropout in the last fully-connected layer with probability $0.5$. We used an ADAM optimizer with base learning rate $\eta = 10^{-4}$. Weights were initialized with the Xavier method. To examine the effect of the network size on learnability, we also repeated our experiments with a larger network control (Fig. 4, purple curve) with 2 times the number of units in the convolution layers and 4 times the number of units in the fully-connected layers.

**Results:** In all conditions, we found a strong dichotomy in the observed learning curves. In conditions where learning occurs (accuracy reached $95\%$), training accuracy suddenly shoots up from chance-level and then gradually approaches $100\%$ accuracy. We call this sudden, dramatic rise in accuracy from chance-level the "learning event". Although we have observed some variation in the point at which the learning event takes place and the rate at which $95\%$ accuracy is eventually reached, the training runs that exhibited a learning event also almost invariably reached $95\%$ accuracy within 20 million training images. On the other hand, when $95\%$ accuracy was never reached, a learning event also almost never took place over the entire length of training. Thus, the final accuracy over different experimental conditions exhibited a strong bi-modality – chance-level or close to $100\%$. In Fig. 4, we report minimum TTA in each experimental condition over the 10 random initializations.

In SR, we found no straining effect across all image parameters over all random initializations. The learning event took place immediately after training begins and accuracy reached $95\%$ soon thereafter. In SD, however, we found a significant straining effect from two image parameters, image size ($n$) and number of items ($k$). For example, increasing image size progressively increases TTA *while also making the learning event less likely*. As a result, the network learned SD in 7 out

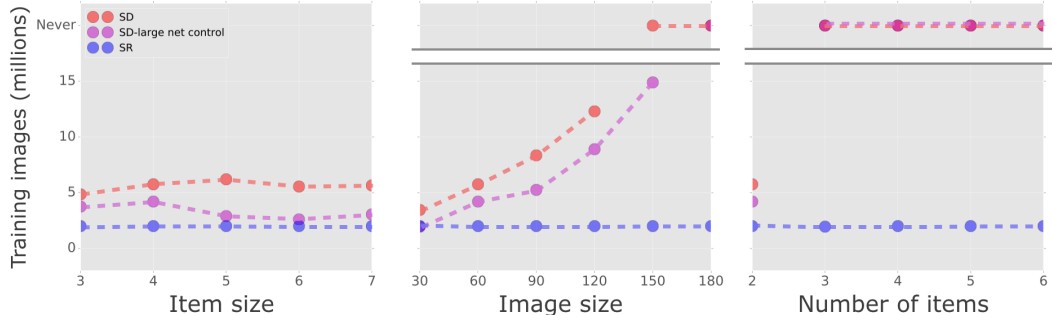

Figure 4: *Training-to-acquisition (TTA) curves over PSVRT image parameters.* TTA denotes the number of training examples needed for a CNN to reach 95% training accuracy. Training used 20 million images, and if accuracy never reached 95% in all of 10 random initializations we consider the problem "never" learned. The figures only show the minimum TTAs out of 10 random initializations in each condition. Three curves – SD (red), SD with a large CNN, (purple) and SR (blue) – are plotted. The three figures display TTA curves over each of three image variability parameters: item size *(a)*, image size *(b)* and number of items *(c)*.

of 10 random initializations for the baseline parameter configuration while it only learned it in 4 out of 10 on $120 \times 120$ images. At image size $150 \times 150$ and above, the network never exhibited a learning event and thus never learned the problem. Increasing the number of items produced an even stronger straining effect. The network never learned the problem when there were 3 or more items in an image.

Additionally, we repeated all three sub-experiments for SD while considering items to be the same if they were congruent up to rotation by a multiple of 90°. This relaxation of the strict same-different rule effectively quadruples the number of matching images in the data set. The CNN never learned for any parameter configuration. Even though this rule is technically a relaxation of the strict same-different rule, the concomitant increase in the number of "same" templates imposes a severe strain on CNNs.

We hypothesize that these straining effects reflect the way these two parameters, image size and item number, contribute to image variability. As we have shown above, image variability is an exponential function of image size as the base and number of items as the exponent. Thus, increasing image size while keeping the number of items as 2 results in a quadratic-rate increase in image variability, while increasing the number of items leads to an exponential-rate increase in image variability. The straining effect was equally strong between two CNNs with more than a twofold difference in network width, only with a constant rightward shift in the TTA curve over image sizes; the transition to the problem being essentially impossible was only delayed by one step in the image size parameter.

In contrast, increasing item size produced no visible straining effect on CNN. Similar to SR, learnability is preserved and stable over the range of item sizes we considered. We realize that it is possible to construct feedforward feature detectors that can generalize to coordinated item variability (i.e., the bit patterns themselves can vary arbitrarily but they always vary together) such as "subtraction templates" with distinct excitatory and inhibitory regions in a particular spatial arrangement, although we have yet to find more direct supporting evidence for such features actually being learned via training in SD.

Taken together, these results imply that, when CNNs learn a PSVRT condition, they are simply building a feature set tailored for a particular data set, instead of learning the "rule" per se. If a network is able to learn features that capture the visual relation at hand (e.g., a feature set to detect *any* pair of items arranged horizontally), then these features should, by definition, be minimally sensitive to the image variations that are irrelevant to the relation. Yet, the CNN in this experiment suffered increasing TTA for increasing image variability, until it simply did not learn at all within our allotted number of training examples. This suggests that the features learned by CNN are not invariant rule-detectors, but rather merely a collection of templates covering a particular distribution in the image space.

## 4 EXPERIMENT 3: RELATIONAL NETWORKS

**The Relational Network:**   Recently, Santoro et al. (2017) proposed the relational network (RN), an architecture explicitly designed to detect visual relations, and tested it on several VQA tasks. This simple feedforward network sits on top of a CNN and learns a map from pairs of high-level CNN feature vectors to the answers to relational questions. Relational questions are either provided to the model as natural language which is then processed by a long short-term memory (LSTM) or simply as hardcoded binary strings. The entire system (CNN+LSTM+RN) can be trained end-to-end. The approach was found to substantially outperform a baseline CNN on various visual reasoning problems.

**Sort-of-CLEVR and its limitations:**   In particular, an RN was able beat a CNN on "sort-of-CLEVR", a VQA task using images with with simple 2D items (Fig. 5a). Scenes had up to six items, each of which had one of two shapes (circle or square) and six colors (yellow, green, orange, blue, red, gray). The RN was trained to answer both relational questions (e.g., *"What is the shape of the object that is farthest from the gray object?"*) and non-relational questions (e.g., *"Is the blue object on the top or bottom of the scene?"*).

The sort-of-CLEVR tasks suffers from two key shortcomings. First, although solving the task requires comparing the attributes of cued items, it does not necessitate learning the concept of sameness per se (*"Are any two items the same in this scene?"*). Second, there are only twelve possible items (2 shapes × 6 colors). Low item variability encourages the RN to solve relational problems using rote memorization of all possible item configurations. In order to understand how RNs perform when these handicaps are removed, we trained the model on both a two-item sort-of-CLEVR same-different task and on PSVRT stimuli. In the former task, our goal was to measure the ability of an RN to transfer the concept of same-different from a training set to a novel set of objects, a classic and very well-studied paradigm in animal psychology (Pepperberg, 1987; Wright and Katz, 2006; Martinho III and Kacelnik, 2016; Wright and Kelly, 2017), and thus an important benchmark for models of visual reasoning.

**Architecture details:**   We used software for relational networks publicly available at `https://github.com/gitlimlab/Relation-Network-Tensorflow`. The convolutional network component of the model had four convolutional layers with kernel sizes of $5 \times 5$ with ReLu activations but no intermittent pooling. The stride was set to 3 in the first two convolutional layers and 2 in the second two. There were 24 features per layer. The RN part of the system comprised a 4-layer MLP with 256 units per layer followed by a 3-layer MLP with 256 units per layer. All fully connected layers in the system except for the last one used ReLu activations. The penultimate layer was trained with 50% dropout. The output of the final layer was passed through a softmax function and the whole system was trained with a cross-entropy loss with an ADAM optimizer with base learning rate $2.5 \times 10^{-4}$. Weights were initialized using Xavier initialization. This is essentially the exact architecture and training procedure used by the original authors (though they did not provide kernel sizes or strides) and we confirmed that this model was able to reproduce the results from (Santoro et al., 2017) on the sort-of-CLEVR task.

**Results:**   We constructed twelve different versions of the sort-of-CLEVR dataset, each one missing one of the twelve possible color+shape combinations. Images in each dataset only depicted two items. Half of the time, these items were the same (same color and same shape). For each dataset, we trained our CNN+RN architecture to detect the possible sameness of the two scene items while measuring validation accuracy on the left-out images. Learning terminated when the model reached 95% training accuracy. We then averaged training accuracy and validation accuracy across all of the left-out conditions.

We found that the CNN+RN does not generalize on average to left-out color+shape combinations on the sort-of-CLEVR task (Fig. 5b). Since there are only 11 color+shape combinations in any given setup, the model does not need to learn to generalize across many items and therefore learns orders of magnitude faster than CNNs on PSVRT stimuli. However, while the average training accuracy curve (solid red) rises rapidly to around 90%, the average validation accuracy remains at chance. In other words, on average, there is no transfer of same-different ability to the left-out condition, even

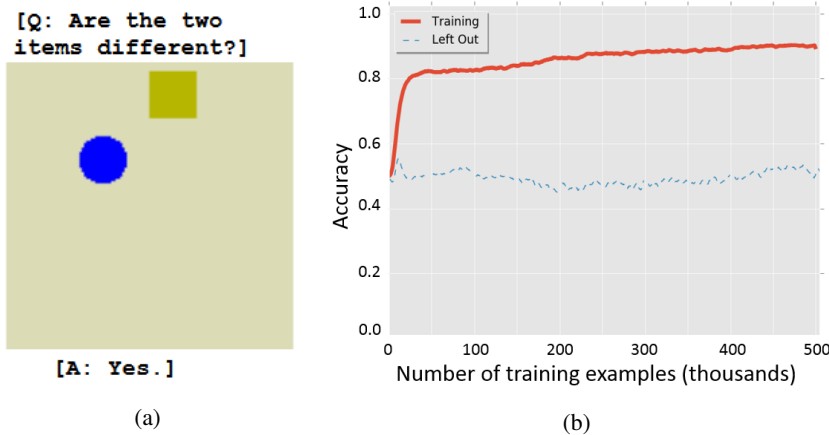

(a)                                              (b)

Figure 5: *(a)* An example of two-item same-different problem posed for a sort-of-CLEVR image. *(b)* Accuracy curves of an RN while being trained in two-item same-different problem on sort-of-CLEVR dataset with one of twelve (2 shapes × 6 colors) item types left out. The red curve shows the accuracy on validation data generated using the same set of items used for training. The blue dashed line shows the accuracy on validation data generated using the left-out items.

though the attributes from that condition (e.g., blue square) were represented in the training set, just not in that combination (e.g., blue circle and red square).

Next, we replaced the simple shapes of sort-of-CLEVR with PSVRT bit patterns. As in Experiment 2, we varied image size from 30 to 180 pixels in increments of 30 while measuring TTA. We trained on 20M images. We repeated training over three different runs to make sure results were stable. We found that the combined CNN+RN behaves essentially like a vanilla CNN. After a long period at chance-level performance over several million images, the CNN+RN leaps to greater than 95% accuracy as long as the image size is 120 or below. For image sizes of 150 and 180, the system did not learn. We speculate that this cutoff point corresponds to the representational capacity of our particular RN architecture. Although we demonstrated this capacity was sufficient to solve the original sort-of-CLEVR task, it is clearly not enough for some same-different tasks on PSVRT.

## 5 DISCUSSION

Our results indicate that visual-relation problems can quickly exceed the representational capacity of CNNs. While learning templates for individual objects appears to be quite tractable for today's deep networks, learning templates for *arrangements* of objects become rapidly intractable because of the combinatorial explosion in the number of templates needed. That stimuli with a combinatorial structure are difficult to represent with feedforward networks has been long acknowledged by cognitive scientists at least as early as Fodor and Pylyshyn (1988). However, this limitation seems to have been somehow overlooked by current computer vision scientists.

Compared to the feedforward networks in this study, biological visual systems excel at detecting relations. Fleuret et al. (2011) found that humans are capable of learning rather complicated visual rules and generalizing them to new instances from just a few SVRT training examples. For instance, their participants could learn the rule underlying the hardest SVRT problem for CNNs in our Experiment 1, problem 20, from an average of about 6 examples. Moreover, problem 20 is rather complicated, involving two shapes such that "one shape can be obtained from the other by reflection around the perpendicular bisector of the line joining their centers." In contrast, the best performing network for this problem in our high-throughput search could not get significantly above chance after a million training examples.

Visual reasoning ability is not just found in humans. For example, birds and primates can be trained to recognize same-different relations and then transfer this knowledge to novel objects (Wright and Katz, 2006). A recent, striking example of same-different learning in animals comes from Martinho III and Kacelnik (2016) who essentially showed that ducklings can perform a one-shot version of

our Experiment 3 from birth. During a training phase, newly hatched ducklings were exposed to a single pair of simple 3D objects that were either the same or different. Later, they demonstrated a preference for novel objects obeying the relationship observed in the training phase. This result suggests that these animals can either rapidly learn the abstract concepts of same and different from a single example or they simply possess these concepts innately. Contrast the behavior of these ducklings with the CNN+RN of Experiment 3, which demonstrated no ability to transfer the concept of same-different to novel objects (Fig. 5b) even after hundreds of thousands of training examples. For a recent review of similar literature (including additional evidence for abstract relational reasoning in pigeons and nutcrackers), see Wright and Kelly (2017).

There is substantial evidence that the neural substrate of visual-relation detection may depend on re-entrant/feedback signals beyond feedforward, pre-attentive processes. It is relatively well accepted that, despite the widespread presence of feedback connections in our visual cortex, certain visual recognition tasks, including the detection of natural object categories, are possible in the near absence of cortical feedback – based primarily on a single feedforward sweep of activity through our visual cortex (Serre, 2016). However, psychophysical evidence suggests that this feedforward sweep is too spatially coarse to localize objects even when they can be recognized (Evans and Treisman, 2005). The implication is that object localization in clutter requires attention (Zhang et al., 2011).

It is difficult to imagine how one could recognize the spatial relation between two objects without spatial information. Indeed, converging neuroscience evidence (Logan, 1994; Moore et al., 1994; Rosielle et al., 2002; Holcombe et al., 2011; Franconeri et al., 2012; van der Ham et al., 2012) suggests that the processing of spatial relations between pairs of objects in a cluttered scene requires attention, even when participants are able to detect the presence of the individual objects pre-attentively, presumably in a single feedforward sweep.

Another brain mechanism that has been implicated in our ability to process visual relations is working memory (Kroger et al., 2002; Golde et al., 2010; Clevenger and Hummel, 2014; Brady and Alvarez, 2015). In particular, imaging studies (Kroger et al., 2002; Golde et al., 2010) have highlighted the role of working memory in prefrontal and premotor cortices when participants solve Raven's progressive matrices which require both spatial and same-different reasoning.

What is the computational role of attention and working memory in the detection of visual relations? One assumption (Franconeri et al., 2012) is that these two mechanisms allow flexible representations of relations to be constructed *dynamically* at run-time via a sequence of attention shifts rather than *statically* by storing visual-relation templates in synaptic weights (as done in feedforward neural networks). Such representations built "on-the-fly" circumvent the combinatorial explosion associated with the storage of templates for all possible relations, helping to prevent the capacity overload associated with feedforward neural networks.

Humans can easily detect when two objects are the same up to some transformation (Shepard and Metzler, 1971) or when objects exist in a given spatial relation (Fleuret et al., 2011; Franconeri et al., 2012). More generally, humans can effortlessly construct an unbounded set of structured descriptions about the visual world around them (Geman et al., 2015). Given the vast superiority of humans over modern computers in their ability to detect visual relations, we see the exploration of attentional and mnemonic mechanisms as an important step in our computational understanding of visual reasoning.

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
