# OpenReview forum: "Not-So-CLEVR: Visual Relations Strain Feedforward Neural Networks"
_ICLR.cc/2018/Conference — Invite to Workshop Track_

### Official Review · AnonReviewer2 · 2017-11-24
**Important limitations of relational networks**

**Rating:** 6
**Confidence:** 3

**Review:**

Quality

This paper demonstrates that convolutional and relational neural networks fail to solve visual relation problems by training networks on artificially generated visual relation data. This points at important limitations of current neural network architectures where architectures depend mainly on rote memorization.

Clarity

The rationale in the paper is straightforward. I do think that breakdown of networks by testing on increasing image variability is expected given that there is no reason that networks should generalize well to parts of input space that were never encountered before.

Originality

While others have pointed out limitations before, this paper considers relational networks for the first time.

Significance

This work demonstrates failures of relational networks on relational tasks, which is an important message. At the same time, no new architectures are presented to address these limitations.

Pros

Important message about network limitations.

Cons

Straightforward testing of network performance on specific visual relation tasks. No new theory development. Conclusions drawn by testing on out of sample data may not be completely valid.

---

> ### Author Response · Authors · 2017-12-18
> **Response to Reviewer 3**
>
> "The rationale in the paper is straightforward. I do think that breakdown of networks by testing on increasing image variability is expected given that there is no reason that networks should generalize well to parts of input space that were never encountered before."
>
> >>Please see the comment entitled "Thank You and Important Clarification". To repeat, in Experiment 2 we *do not* test the CNNs on new image scenarios after training on other image conditions. For each setting of image parameters, a network is trained and tested from scratch on the same data set to obtain a TTA.  Each dot in Figure 4 represents a repetition of this procedure for a new data distribution defined by different item size, image size and item number parameters. The purpose of the experiment is to measure how TTA is affected by image variability. Roughly, if the CNN learned the “rule”, then TTA should not have increased with image variability, since all images obey the rule, regardless of the image parameters. If it can only ‘seem to solve it’ by fitting to a particular image distribution, then we would expect TTA to increase with increasing image variability. The confusion may have arisen because of our erroneous use of the term ‘generalization’ in Experiment 2 which, in machine learning literature, refers to the ability to explain new data given a fixed training dataset. We have revised the manuscript to reduce confusion.
>
> "Straightforward testing of network performance on specific visual relation tasks. No new theory development. Conclusions drawn by testing on out of sample data may not be completely valid."
>
> >>Regarding the reviewer’s criticism about out-of-sample data, we would like to clarify once again that all testing data was in-sample except in Experiment 3. When we actually do use out of sample data, in the CNN+RN experiment, we do so in a way that animals are known to solve.  For example, we cite a study by Martinho and Kacelnik (2016) in Science showing that ducklings, via imprinting, can learn as well as generalize same-different visual relations immediately after birth. During a training phase, ducklings were exposed to a single pair of simple 3D objects that were either the same or different. Later, they demonstrated a preference for novel objects obeying the relationship observed in the training phase. The conclusion of the authors is that these animals can either rapidly learn the abstract concepts of same and different from a single example or they simply possess these concepts innately. For a recent review of similar literature (including additional evidence for abstract relational reasoning ability in pigeons and nutcrackers), see Wright and Kelly (2017) in Learning and Behavior. Our main theoretical contribution was the first systematic analysis of CNNs on visual-relation problems, varying network hyperparameters and image parameters to show that some visual relations are qualitatively harder than others (Experiment 1,2). We showed that this difference is due neither to a particular architectural choice (Experiment 1) nor to factors unrelated to the visual relations themselves such as image distribution (Experiment 2). In Experiments 2 and 3, we demonstrate that CNNs are limited in their ability to learn and represent abstract rules underlying same-different relations, and instead only solve it by rote memorization. We contrast these results with biological vision, where mechanisms other than template matching play a critical role in learning and detecting visual relations.

---

### Official Review · AnonReviewer1 · 2017-11-26
**review of "NOT-SO-CLEVR: VISUAL RELATIONS STRAIN FEEDFORWARD NEURAL NETWORKS"**

**Rating:** 6
**Confidence:** 3

**Review:**

The authors introduce a set of very simple tasks that are meant to illustrate the challenges of learning visual relations.  They then evaluate several existing network architectures on these tasks, and show that results are not as impressive as others might have assumed they would be.   They show that while recent approaches (e.g. relational networks) can generalize reasonably well on some tasks, these results do not generalize as well to held-out-object scenarios as might have been assumed.

Clarity:  The paper is fairly clearly written.   I think I mostly followed it.

Quality:  I'm intrigued by but a little uncomfortable with the generalization metrics that the authors use.   The authors estimate the performance of algorithms by how well they generalize to new image scenarios when trained on other image conditions.   The authors state that ". . . the effectiveness of an architecture to learn visual-relation problems should be measured in terms of generalization over multiple variants of the same problem, not over multiple splits of the same dataset."  Taken literally, this would rule out a lot of modern machine learning, even obviously very good work. On the other hand, it's clear that at some point, generalization needs to occur in testing ability to understand relationships.  I'm a little worried that it's "in the eye of the beholder" whether a given generalization should be expected to work or not.

There are essentially three scenarios of generalization discussed in the paper:
        (a) various generalizations of image parameters in the PSVRT dataset
        (b) various hold-outs of the image parameters in the sort-of-CLEVR dataset
        (c) from sort-of-CLEVR "objects" to PSVRT bit patterns

The result that existing architectures didn't do very well at these generalizations (especially b and c) *may* be important -- or it may not.    Perhaps if CNN+RN were trained on a quite rich real-world training set with a variety of real-world three-D objects beyond those shown in sort-of-CLEVR, it would generalize to most other situations that might be encountered.    After all, when we humans generalize to understanding relationships, exactly what variability is present in our "training sets" as compared to our "testing" situations?   How do the authors know that humans are effectively generalizing rather than just "interpolating" within their (very rich) training set?  It's not totally clear to me that if totally naive humans (who had never seen spatial relationships before) were evaluated on exactly the training/testing scenarios described above, that they would generalize particularly well either.   I don't think it can just be assumed a priori that humans would be super good this form of generalization.

So how should authors handle this criticism?  What would be useful would either be some form of positive control.  Either human training data showing very effective generalization (if one could somehow make "novel" relationships unfamiliar to humans), or a different network architecture that was obviously superior in generalization to CNN+RN. If such were present, I'd rate this paper significantly higher.

Also, I can't tell if I really fully believe the results of this paper.  I don't doubt that the authors saw the results they report.  However, I think there's some chance that if the same tasks were in the hands of people who *wanted* CNNs or CNN+RN to work well, the results might have been different.   I can't point to exactly what would have to be different to make things "work", because it's really hard to do that ahead of actually trying to do the work.   However, this suspicion on my part is actually a reason I think it might be *good* for this paper to be published at ICLR.  This will give the people working on (e.g.) CNN+RN somewhat more incentive to try out the current paper's benchmarks and either improve their architecture or show that the the existing one would have totally worked if only tried correctly.  I myself am very curious about what would happen and would love to see this exchange catalyzed.

Originality and Significance:  The area of relation extraction seems to me to be very important and probably a bit less intensively worked on that it should be.  However, as the authors here note, there's been some recent work (e.g. Santoro 2017) in the area.   I think that the introduction of baselines  benchmark challenge datasets such as the ones the authors describe here is very useful, and is a somewhat novel contribution.

---

> ### Author Response · Authors · 2017-12-18
> **Response to Reviewer 2**
>
> "I'm intrigued by but a little uncomfortable with the generalization metrics that the authors use... "
>
> >> Please see the comment entitled "Thank You and Important Clarification". In short, we *do not* test generalization in Experiments 1 or 2.
>
> "... I don't think it can just be assumed a priori that humans would be super good this form of generalization."
>
> >> Although we are not aware of an experiment done on human infants learning same-different, we do cite a study by Martinho and Kacelnik (2016) showing that ducklings can learn same-different visual relations immediately after birth. During a training phase, newly-hatched ducklings were exposed to a single pair of 3D objects that were either the same or different. Later, they preferred novel objects obeying the relationship observed in the training phase. The conclusion is that these animals can either rapidly learn the abstract concepts of same and different from a single example or they simply possess these concepts innately. For a recent review of similar literature, see Wright and Kelly (2017). Our experiment 3 is essentially analogous to this. Taken in conjunction with the results from Experiment 2, we conclude that state-of-the-art feedforward architectures only learn same-different relation via memorization of examples. We have expanded the discussion to include these points.
>
> "What would be useful would either be some form of positive control. Either human training data showing very effective generalization...or a different network architecture that was obviously superior in generalization to CNN+RN."
>
> >>While there is a substantial literature specifically on same-different detection in humans going back to Donderi & Zelnicker (1969), the only experiment known to us in which humans are tested on many relation problems is Fleuret et al., 2011. The authors found that humans can learn rather complicated visual rules and generalize them to new instances from just a few examples. Their subjects could learn the rule underlying SVRT problem 20 (the hardest problem for CNNs in our Experiment 1) from about 6 examples. Problem 20 was a complicated problem, involving two shapes such that “one shape can be obtained from the other by reflection around the perpendicular bisector of the line joining their centers.” (See revised manuscript, Discussion, paragraph 2). While there is currently no model with superior generalization compared to a CNN+RN, Ellis et al. (2015) found program synthesis could vastly outperform two different CNN architectures on SVRT. Still, the best visual reasoning machine we know of is the human brain, which is why we suggest attention and memory as the solution to our visual-relation challenges.
>
> "However, I think there's some chance that if the same tasks were in the hands of people who *wanted* CNNs or CNN+RN to work well, the results might have been different."
>
> >> We agree with the reviewer that special care must be taken in criticizing any model. But, note that we do not argue that it is absolutely impossible for *some* CNN to solve a given visual-relation problem. This absolute claim must be false, since feedforward networks are universal function approximators. Rather, the final argument we make in this paper is a relative one: some visual relations are harder than others for CNNs. To support this, we relied on properties diagnostic of rote memorization (e.g. sensitivity to network size in Experiment 1 and sensitivity to image variability in Experiment 2) that are present in same-different results and not in spatial relations results. We varied the CNN architecture (Experiment 1) and image parameters (Experiment 2) to ensure that the qualitative differences between the results obtained from spatial relations and same-different relations are neither due to a particular image distribution nor to a particular CNN hyperparameter choice.  We would also like to reassure the reviewers that the experiments were designed with little room for manipulation. The hyperparameters we chose for the CNNs were well within the ‘standard’ range in the CNN literature. Additionally, in Experiment 2 we first chose baseline image parameters and CNN hyperparameters that ensure a very low TTA for both SR and SD problems. Then we simply repeated the training while varying each parameter. We are confident that the trend we observed here will hold outside the range of hyperparameters we have considered. Further, we believe that the limitations of feedforward networks on visual-relation problems have already been recognized by the machine learning community, who have begun to use models based on program induction and memory (e.g., “Inferring and Executing Programs for Visual Reasoning,” Johnson et al., 2017, ICCV). We hope that the challenges we pose in this paper can be used as a benchmark for these new models.

---

### Official Review · AnonReviewer3 · 2017-11-28
**This paper explores how current CNN’s and Relational Networks fail to recognize visual relations in images. It firstly performs a performance analysis of CNN’s on SVRT, proposes a new visual relation challenge and shows how the proposed sort-of-CLEVR challenge can be slightly modified to break current state of the art approaches.**

**Rating:** 6
**Confidence:** 4

**Review:**

Strengths:

-	There is an interesting analysis on how CNN’s perform better Spatial-Relation problems in contrast to Same-Different problems, and how Spatial-Relation problems are less sensitive to hyper parameters.

-	The authors bring a good point on the limitations of the SVRT dataset – mainly being the difficulty to compare visual relations due to the difference of image structures on the different relational tasks and the use of simple closed curves to characterize the relations, which make it difficult to quantify the effect of image variability on the task. And propose a challenge that addresses these issues and allows controlling different aspects of image variability.

-	The paper shows how state of the art relational networks, performing well on multiple relational tasks, fail to generalize to same-ness relationships.

Weaknesses:

-	While the proposed PSVRT dataset addresses the 2 noted problems in SVRT, using only 2 relations in the study is very limited.

-	The paper describes two sets of relationships, but it soon suggests that current approaches actually struggle in Same-Different relationships. However, they only explore this relationship under identical objects. It would have been interesting to study more kinds of such relationships, such as equality up to translation or rotation, to understand the limitation of such networks. Would that allow improving generalization to varying item or image sizes?

Comments:

-	In page 2, authors suggest that from that Gülçehre, Bengio (2013) that for visual relations “failure of feed-forward networks […] reflects a poor choice of hyper parameters. This seems to contradict the later discussion, where they suggest that probably current architectures cannot handle such visual relationships.

-	The point brought about CNN’s failing to generalize on same-ness relationships on sort-of-CLEVR is interesting, but it would be good to know why PSVRT provides better generalization. What would happen if shapes different than random squared patterns were used at test time?

-	Authors reason about biological inspired approaches, using Attention and Memory, based on existing literature. While they provide some good references to support this statement it would have been interesting to show whether they actually improve TTA under image parameter variations

---

> ### Author Response · Authors · 2017-12-18
> **Response to Reviewer 1**
>
> "While the proposed PSVRT dataset addresses the 2 noted problems in SVRT, using only 2 relations in the study is very limited."
>
> >> We agree. Although it would certainly be interesting to extend this investigation to a larger set of relations, we limited our focus to these two relations because 1)  we wanted to ensure that the relations are defined on the same image distributions, and it is not easy to satisfy this requirement if we include other visual relations, 2) we believe that relative position and sameness are the key factors underlying the dichotomy of CNN accuracies in Experiment 1, and 3) In human and animal psychology, the detection of horizontal/vertical relations and of sameness/difference is a well-established protocol, so using these two PSVRT problems will make it easy to eventually collect human data.
>
> "The paper describes two sets of relationships, but it soon suggests that current approaches actually struggle in Same-Different relationships. However, they only explore this relationship under identical objects. It would have been interesting to study more kinds of such relationships, such as equality up to translation or rotation, to understand the limitation of such networks. Would that allow improving generalization to varying item or image sizes?"
>
> >> First of all, please see our above note about generalization. To repeat, our PSVRT task does not measure generalization to left-out regions of the input space . Figure 4 reports the number of samples required to achieve 95% accuracy on the training set. There was no holdout set. Second, we were very intrigued by your suggestion to include rotated items in our same-different experiment. We hypothesized that, as including rotations simply increases the number of ways that items can be “the same,” sample complexity would actually be worse than the PSVRT same-different without rotations. The paper is now updated to show the results of this test. Indeed, the baseline CNN architecture never learned for any parameter configuration on this new task.
>
> "In page 2, authors suggest that from that Gülçehre, Bengio (2013) that for visual relations “failure of feed-forward networks […] reflects a poor choice of hyper parameters. This seems to contradict the later discussion, where they suggest that probably current architectures cannot handle such visual relationships. "
>
> >> The citation was made in order to acknowledge the possibility that such previous demonstrations as Gülçehre and Bengio (2013) could have simply reflected poor hyperparameter choices. From this, we motivate our experimental paradigm (Experiment 1) where we used 9 different architectures with varying filter sizes and depth. We found that hyperparameters made little difference on the ‘difficult’ problems, with less than 10% difference in final accuracy between the worst-case and the best-case (Page 3). We have added a sentence at the end of Experiment 1 to make that point clearer.
>
> "The point brought about CNN’s failing to generalize on same-ness relationships on sort-of-CLEVR is interesting, but it would be good to know why PSVRT provides better generalization. What would happen if shapes different than random squared patterns were used at test time?"
>
> >> Again, please see our opening remarks. Only training accuracy was measured for PSVRT and there was no holdout test set. Testing accuracy on a left-out condition was only measured in Experiment 3, with CNN+RN on sort-of-CLEVR dataset. However, the referee’s question inspired us to measure generalization in earnest on PSVRT by training a network to high accuracy on one problem with one parameter configuration and then testing it on all other parameter settings. Just as the referee suggests, test accuracy monotonically decreases as the image parameters begin to deviate from their training settings. This decrease was always sharper in same-different problems than in spatial problems.
>
> "Authors reason about biological inspired approaches, using Attention and Memory, based on existing literature. While they provide some good references to support this statement it would have been interesting to show whether they actually improve TTA under image parameter variations"
>
> >> Our goal with this paper was to systematically probe the limits of feedforward networks on visual relations problems. We believe our analysis is thorough and fits nicely within the space constraints of a conference paper.

---

### Public Comment · (anonymous) · 2017-12-18
**Reference missing**

The general idea and specially the first experiment (using Fleuret's stimuli) is quite similar to a work published last year at ICANN: https://arxiv.org/pdf/1607.08366.pdf
I think that paper should at least be cited.

---

> ### Author Response · Authors · 2017-12-18
> **Citation added to the paper**
>
> Thank you for bringing this reference to our attention. An updated submission will include this citation.

---

### Author Response · Authors · 2017-12-18
**Thank you and Important Clarification**

First, we would like to thank the reviewers for their thoughtful comments, which we believe strengthen the paper greatly.

Second, we would like to make a general clarification regarding all three experiments we ran. Except in our third experiment, we do not test for network generalization to new regions of the input space. We believe that the original description of our experiments was unclear, and our use of the word “generalization” to describe the behavior of CNNs on PSVRT was erroneous. The manuscript has been revised accordingly.

In the PSVRT experiment, for example, the TTA obtained in each condition denotes the number of training examples required for a CNN to obtain 95% validation accuracy on images sampled from the same image distribution as the training images. This procedure was replicated over multiple image parameter configurations, resulting in TTAs as shown in Figure 4. There was no holdout set with a different image distribution than the training set. The purpose of the experiment is to measure how TTA was affected by image variability. If a CNN could learn the “rule”, then TTA should not have increased with image variability, since all images obey the rule, regardless of the image parameters. But in our experiment, TTA increased.

Only in the third experiment do we test a network (the CNN+RN) on images with combinations of attributes not in the training set. However, we now emphasize in the paper that exactly this kind of generalization is indeed found in biological organisms, essentially from birth (see revised manuscript, Discussion, paragraph 3).

The following is the exhaustive list of revisions we made to the manuscript and where readers can find them:
1. In Results in Experiment 1 (SVRT) we added a reference to Stabinger et al. (2016).
2. We changed the Method and architectural details in Experiment 2 (PSVRT) to make it clearer that a CNN is not tested for generalization but instead is trained from scratch for each image parameter to obtain the TTA curves.
3. In Experiment 3 (RN on Sort-of-CLEVR) and Discussion we emphasized with additional citations the fact that animals are capable of the kinds of generalization we tested using Sort-of-CLEVR.
4. In Results in Experiment 2 (PSVRT) we added the result from another task, same-different up to rotation, as a reviewer requested.
5. Minor edits for clarity and to correct typos.

---

### Decision · Program_Chairs · 2018-01-29
**ICLR 2018 Conference Acceptance Decision**

**Decision:**

Invite to Workshop Track

**Comment:**

This paper studies an important problem (visual relationship detection and generalization capabilities existing networks for this task). Unfortunately, all reviewers raise concerns (e.g. limited relations studied) and are largely on the fence about this paper. While this paper does not propose solutions, it does present interesting "negative results" that should get some visibility in the workshop track.